# MixEdit: Revisiting Data Augmentation and Beyond for Grammatical Error Correction

**Jingheng Ye[1], Yinghui Li[1*], Yangning Li[1,2], Hai-Tao Zheng[1,2*]**
[1]Tsinghua Shenzhen International Graduate School, Tsinghua University
[2]Peng Cheng Laboratory
{yejh22,liyinghu20}@mails.tsinghua.edu.cn

## Abstract

Data Augmentation through generating pseudo data has been proven effective in mitigating the challenge of data scarcity in the field of Grammatical Error Correction (GEC). Various augmentation strategies have been widely explored, most of which are motivated by two heuristics, i.e., increasing the distribution similarity and diversity of pseudo data. However, the underlying mechanism responsible for the effectiveness of these strategies remains poorly understood. In this paper, we aim to clarify how data augmentation improves GEC models. To this end, we introduce two interpretable and computationally efficient measures: Affinity and Diversity. Our findings indicate that an excellent GEC data augmentation strategy characterized by high Affinity and appropriate Diversity can better improve the performance of GEC models. Based on this observation, we propose MixEdit, a data augmentation approach that *strategically* and *dynamically* augments realistic data, without requiring extra monolingual corpora. To verify the correctness of our findings and the effectiveness of the proposed MixEdit, we conduct experiments on mainstream English and Chinese GEC datasets. The results show that MixEdit substantially improves GEC models and is complementary to traditional data augmentation methods [1].

## 1 Introduction

Grammatical Error Correction (GEC) is a task that involves making *locally substitutions* in text to correct all grammatical errors in a text (Ye et al., 2023b; Li et al., 2023; Ye et al., 2023a; Ma et al., 2023). GEC is often considered a monolingual machine translation (MT) task, which is typically tackled using sequence-to-sequence (Seq2Seq) architecture (Bryant et al., 2022; Dong et al., 2023).

Numerous studies tend to improve the performance of Seq2Seq GEC models by increasing the amount of training data (Ye et al., 2022; Li et al., 2022c,b; Zhang et al., 2023).

However, high-quality parallel data for GEC is not as widely available. Despite the great success of Seq2Seq models, they are prone to overfitting and making predictions based on spurious patterns (Tu et al., 2020), owing to the vast gap between the number of model parameters and the limited high-quality data. This data sparsity issue has motivated research into data augmentation in the field of GEC, particularly in the context of resource-heavy Seq2Seq approaches (Rothe et al., 2021; Stahlberg and Kumar, 2021). Thanks for the ease of constructing pseudo grammatical errors, recent studies focus on generating synthetic parallel data from clean monolingual corpora. Various augmentation methods have been widely explored, whose major motivation comes from improving 1) **distribution similarity** between pseudo and realistic data, including *error patterns* (Choe et al., 2019) and *back-translation* (Sennrich et al., 2016a; Xie et al., 2018; Stahlberg and Kumar, 2021); and 2) **diversity** of pseudo data (Koyama et al., 2021b), such as *noise injection* (Grundkiewicz et al., 2019; Xu et al., 2019) and *round-trip translation* (Zhou et al., 2020). However, prior works have primarily focused on showing the effectiveness of their proposed augmentation methods, without considering sample efficiency. Training GEC models with excessive samples (e.g. over 100M (Lichtarge et al., 2019; Stahlberg and Kumar, 2021)) for poor-scalable improvement is expensive and often unfeasible for most researchers. Additionally, existing studies suffer from a lack of consistent experimental settings, making it intractable to systematically and fairly compare various data augmentation methods. In this complex landscape, claims regarding distribution similarity and diversity remain unverified heuristics. An uncomplicated approach

---

[‡]Corresponding authors: Hai-Tao Zheng and Yinghui Li.

[1]All the source codes of MixEdit are released at https://github.com/THUKElab/MixEdit.

| | |
|---|---|
| **Target** | This will , if not already , cause problems as there is very limited space for us . |
| **Source** | This will , if not already , caused problems as there are very limited spaces for us . |
| **Direct Noise (DN)** | ⟨mask⟩ will , I ⟨mask⟩ already ⟨mask⟩ will cause problems the ⟨mask⟩ is ⟨mask⟩ ⟨mask⟩ space for ⟨mask⟩ . |
| **Pattern Noise (PN)** | This will , if not already , cause problem as there is very limiting space for us ? |
| **Backtranslation (BT)** | This if not already cause the problems as there are very few space for us . |
| **Round-translation (RT)** | If that had not been done , it would have caused problems , because our space is very limited . |
| **Target** | We realize that burning of fuels produces a large amount of greenhouse gases . |
| **Source** | We relize that burning of fuels produce the large amount of greenhouse gases . |
| **Direct Noise (DN)** | We ⟨mask⟩ ⟨mask⟩ burning of looking ⟨mask⟩ a power ⟨mask⟩ ⟨mask⟩ greenhouse gases ⟨mask⟩ |
| **Pattern Noise (PN)** | we realized that burnings of fuels produces a large amount of greenhouse gases . |
| **Backtranslation (BT)** | We realize that burn of fuel produce a lot of greenhouse gases . |
| **Round-translation (RT)** | We recognize that large amounts of greenhouse gases are generated by combustion fuels . |

Table 1: Examples of pseudo source sentences generated by Direct Noise (DN), Pattern Noise (PN), Backtranslation (BT) and Round-translation (RT), respectively.

to evaluate the effectiveness of an augmentation heuristic is to conduct experiments on all possible augmented datasets. However, it is computationally expensive and even impractical when confronted with numerous augmentation heuristic options.

In this paper, to determine the extent to which distributional similarity and diversity of data augmentation can improve GEC models, we quantify both heuristics. We use Affinity measure to evaluate the distribution similarity between pseudo and realistic grammatical errors, which is defined as the inverse of Kullback–Leibler (KL) divergence between them. On the other hand, Diversity measure is used to assess the uncertainty of grammatical errors, defined as the entropy. Next, we revisit four mainstream GEC data augmentation methods using our proposed measures. Our findings illustrate that Affinity varies across these augmentation methods and is positively correlated with the performance of GEC models to some extent. By altering the corruption rate of an augmentation method, we observe that the varying Diversity serves as a trade-off between Precision and Recall.

To overcome the challenge of limiting an augmentation strategy with high Affinity and appropriate Diversity, we propose MixEdit, a data augmentation approach that *strategically* and *dynamically* augments realistic data. Unlike traditional approaches that rely on pseudo data generated from extra monolingual corpora, MixEdit regularizes over-parameterized GEC models using a limited amount of realistic data during fine-tuning. MixEdit achieves this by replacing grammatical errors in the source sentence with other probable and label-preserving grammatical errors, avoiding undesired noise. These augmented samples differ only in the form of grammatical errors, encouraging models to fully utilize the intrinsic informa-

tion among diverse augmented samples, instead of learning spurious patterns during training. We further apply Jensen-Shannon divergence consistency regularization to match the predictions between error patterns, and dynamically select candidate grammatical errors during fine-tuning.

We conduct experiments on two English GEC evaluation datasets: CoNLL-14 (Ng et al., 2014) and BEA-19 (Bryant et al., 2019), and two Chinese GEC evaluation datasets: NLPCC-18 (Zhao et al., 2018) and MuCGEC (Zhang et al., 2022a). Despite its simplicity, MixEdit consistently leads to significant performance gains compared to traditional methods. By combining MixEdit with traditional methods, we achieve state-of-the-art results on BEA-2019, NLPCC-2018, and MuCGEC.

## 2 Background

To avoid confusion, we use the uppercase symbol $X$ to represent a sentence, and the bold lowercase symbol $\boldsymbol{x}$ to indicate a text segment. In this paper, we focus the discussion on constructing grammatical errors for GEC data augmentation. Generally, Seq2Seq-based GEC models parameterized by $\boldsymbol{\theta}$ learn the translation probability $P(Y \mid X; \boldsymbol{\theta})$, where $X$ denotes an ungrammatical source sentence and $Y$ represents a grammatical target sentence. Given a parallel training dataset $\mathcal{D}$, the standard training objective is to minimize the empirical risk:

$$\mathcal{L}(\boldsymbol{\theta}) = \mathbb{E}_{(X,Y)\sim\mathcal{D}}[\mathcal{L}_{\text{CE}}(X, Y; \boldsymbol{\theta})], \quad (1)$$

where $\mathcal{L}_{\text{CE}}$ is the cross entropy loss, $\mathcal{D}$ could be a realistic dataset $\mathcal{D}_r$ in a standard supervised learning setting, or a pseudo dataset $\mathcal{D}_p$ in typical GEC data augmentation settings, where the source sentences are usually generated from monolingual cor-

pora (Kiyono et al., 2020).

Recent works have concentrated on improving the performance of GEC models by integrating various data augmentation techniques, which usually fall under the categories listed in Table 1.

**Direct Noise (DN).** DN injects noise into grammatically correct sentences in a rule-based manner (Kiyono et al., 2020). The noise can take the form of 1) masking, 2) deletion, and 3) insertion based on pre-defined probabilities. DN is applicable to all languages since its rules are language-independent. However, the generated errors are often not genuine and may even distort the original semantics of the sentences.

**Pattern Noise (PN).** PN, on the other hand, involves injecting grammatical errors that are already present in the realistic GEC dataset into sentences (Choe et al., 2019). Specifically, this process entails first identifying error patterns in the GEC dataset using an automated error annotation tool such as ERRANT (Bryant et al., 2017), followed by applying a noising function that randomly substitutes text segments with grammatical errors.

**Backtranslation (BT).** With the help of Seq2Seq models, BT can generate more genuine grammatical errors by learning the distribution of human-written grammatical errors (Xie et al., 2018). The noisy model is trained with the inverse of GEC parallel dataset, where ungrammatical sentence are treated as the target and grammatical ones as the source. Xie et al. (2018) proposed several variants of BT, and showed that the variant **BT (Noisy)** achieved the best performance. Therefore, we focus on this variant in this work. When decoding the ungrammatical sentences, BT (Noisy) adds $r\beta_{\text{random}}$ to the score of each hypothesis in the beam for each time step, where $r$ is drawn uniformly from the interval $[0, 1]$, and $\beta_{\text{random}}$ is a hyper-parameter that controls the degree of noise.

**Round-translation (RT).** RT is an alternative method to generate pseudo data, which is based on the assumption that NMT systems may produce translation errors, resulting in noisy outputs via the bridge languages (Lichtarge et al., 2019; Zhou et al., 2020). The diverse outputs, however, may change the structure of the sentence due to the heterogeneity of different languages.

**Training Settings.** There are two primary training settings for incorporating $\mathcal{D}_p$ into the optimiza-

tion of Equation (1): 1) *jointly optimizing* GEC models by concatenating the realistic dataset $\mathcal{D}_r$ and the pseudo datset $\mathcal{D}_p$, and 2) *pre-training* models on the pseudo dataset before fine-tuning on realistic datasets. In a study conducted by Kiyono et al. (2020), these two training settings were compared for two data augmentation methods (DN and BT), and it was found that pre-training was superior to joint optimization when large enough pseudo data was available. Therefore, to avoid any adverse effects resulting from noisy augmented samples, we adopt the pre-training setting in our work.

## 3 Method

### 3.1 Affinity and Diversity

**Affinity.** In-distribution corruption (Choe et al., 2019) has motivated the design of GEC data augmentation policies (Grundkiewicz et al., 2019; Stahlberg and Kumar, 2021; Yasunaga et al., 2021), based on the idea that pseudo data with less distribution shift should improve performance on specific evaluation datasets. Inspired by this focus, we propose Affinity, which is used to qualify how augmentation shifts data with respect to the error pattern. We define Affinity of a data augmentation method as the inverse of Kullback–Leibler (KL) divergence between pseudo and realistic grammatical errors, which can be computed as follow:

$$\frac{1}{\text{Affinity}(\mathcal{D}_p, \mathcal{D}_r)} = \frac{1}{2}\text{KL}(P_p(\boldsymbol{x}, \boldsymbol{y}) \parallel P_r(\boldsymbol{x}, \boldsymbol{y})) + \frac{1}{2}\text{KL}(P_r(\boldsymbol{x}, \boldsymbol{y}) \parallel P_p(\boldsymbol{x}, \boldsymbol{y})), \quad (2)$$

where $\mathcal{D}_p$ and $\mathcal{D}_r$ refer to the pseudo and realistic datasets, respectively. The pair of text corrections, denoted by $\boldsymbol{x}$ and $\boldsymbol{y}$, can be extracted using an automated error annotation toolkit such as ERRANT (Bryant et al., 2017). $P_p$ and $P_r$ denote the probabilities of pseudo and realistic grammatical errors, and KL represents KL divergence used to calculate the distance between two distributions. To prevent KL approaching infinity, we limit the support set of the calculation. The first term of the above equation is computed as follow:

$$\text{KL}(P_p(\boldsymbol{x}, \boldsymbol{y}) \parallel P_r(\boldsymbol{x}, \boldsymbol{y})) = \sum_{(\boldsymbol{x}, \boldsymbol{y}) \sim \mathcal{D}_r} P_p(\boldsymbol{x}, \boldsymbol{y}) \log \frac{P_p(\boldsymbol{x}, \boldsymbol{y})}{P_r(\boldsymbol{x}, \boldsymbol{y})}. \quad (3)$$

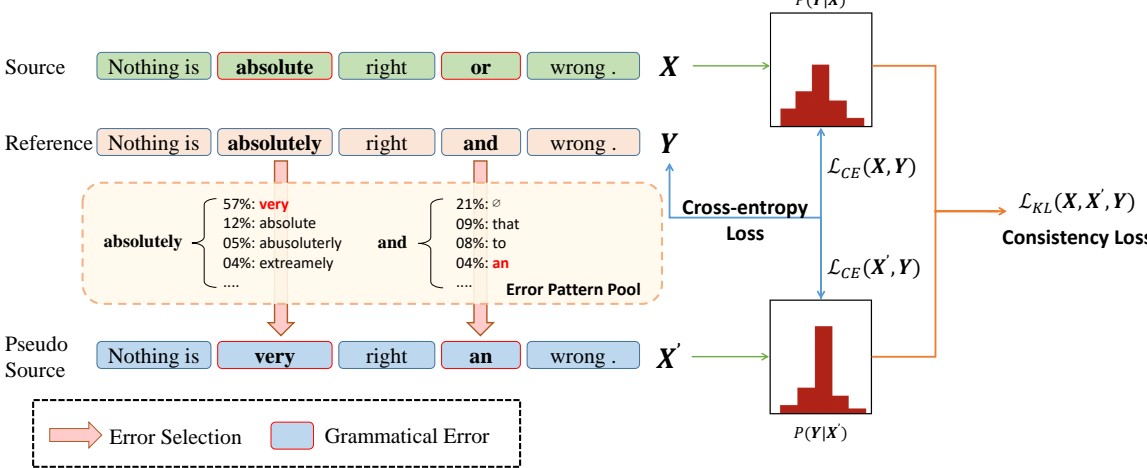

Figure 1: Overview of our approach MixEdit. MixEdit 1) first extracts the error patterns from GEC realistic datasets and builds **Error Pattern Pool**, 2) replaces grammatical errors with alternative candidates from the Error Pattern Pool, and then 3) computes the cross-entropy loss $\mathcal{L}_{\text{CE}}$ and the consistency loss $\mathcal{L}_{\text{KL}}$.

With this definition, Affinity of a small value suggests the pseudo grammatical errors are more likely to be distributed within the realistic dataset.

**Diversity.** Koyama et al. (2021b) demonstrated the importance of diversity in pseudo grammatical errors for improving performance. Based on the observation, we introduce another perspective to evaluate GEC data augmentation policies, which we dub Diversity, defined as the entropy of pseudo grammatical errors:

$$\text{Diversity} = -\sum_{(\boldsymbol{x},\boldsymbol{y})} P(\boldsymbol{x},\boldsymbol{y}) \log P(\boldsymbol{x},\boldsymbol{y}), \quad (4)$$

where $P(\cdot)$ can take on the values of $P_p(\cdot)$ or $P_r(\cdot)$. Like Affinity, this definition of Diversity has the advantage of capturing task-specific and interpretable elements, i.e., the error patterns of GEC datasets. Furthermore, these measures are off-the-shelf and require little computational cost.

It should be kindly noted that the performance of GEC models is not purely a function of training data, as training dynamics and implicit biases in the model can also impact final performance. Both of these measures are introduced to provide a new perspective on characterizing and understanding GEC data augmentation.

### 3.2 MixEdit

MixEdit aims to strategically and dynamically construct pseudo data with high Affinity and appropriate Diversity, thus achieving the best of both worlds. Specifically, we first extract error patterns

from a GEC dataset $\mathcal{D}$. The following corruption probability can be derived from the error patterns using Bayes' rule:

$$P(\boldsymbol{x} \mid \boldsymbol{y}) = \frac{P(\boldsymbol{x},\boldsymbol{y})}{P(\boldsymbol{y})}. \quad (5)$$

The corruption probability describes the correction feature of GEC dataset. For each parallel sample $(X,Y)$, the correction can be denoted by an edit sequence $E = \{\boldsymbol{e}_1, \boldsymbol{e}_2, \cdots, \boldsymbol{e}_m\}$, where each edit $\boldsymbol{e}_i$ consists of a source segment $\boldsymbol{x}$ and a target segment $\boldsymbol{y}$. As shown in Figure 1, MixEdit dynamically replace the original source segment $\boldsymbol{x}$ with other candidates $\boldsymbol{x}'$ from a pool of error patterns during training. Since all the candidates share the same target segment $\boldsymbol{y}$, this transformation is typically label-preserving, preventing undesired noise. These label-preserving perturbations provides various literal forms of the same grammatical errors, enabling correct predictions based on more complex contexts instead of spurious patterns.

Inspired by Shen et al. (2020); Chen et al. (2021), we incorporate consistency regularization in our model to encourage stable and similar predictions across realistic and augmented samples. The training objective is written as: [2]

$$\mathcal{L} = \mathcal{L}_{\text{CE}}(X,Y) + \alpha\mathcal{L}_{\text{CE}}(X',Y) \\ + \beta\mathcal{L}_{\text{KL}}(X,X',Y), \quad (6)$$

---

[2]Shen et al. (2020) considered multiple augmented samples for a realistic sample, but we found a single augmented sample works just as well. Additionally, increasing the number of augmented samples significantly escalates the training cost.

where the augmented sample $X'$ is generated dynamically during training, with weights $\alpha$ and $\beta$ used to balance the contribution of learning from the original data and the augmented data. $\mathcal{L}_{\mathrm{CE}}$ denotes the cross-entropy loss, and $\mathcal{L}_{\mathrm{KL}}$ is KL-divergence as follow:

$$\mathcal{L}_{\mathrm{KL}}(X, X', Y) = \mathrm{KL}\left[P(Y \mid X') \parallel P_{avg}\right], \quad (7)$$

where $P_{avg}$ represents the average prediction probability across realistic and augmented samples.

PN is the most similar traditional GEC data augmentation method to MixEdit. However, there are several distinctions between these two methods: 1) MixEdit provides different views for the same grammatical errors, while PN randomly constructs pseudo errors. In our preliminary experiments [3], we found it matters to determine the positions for constructing grammatical errors. 2) PN is typically used to generate new pseudo parallel data from monolingual corpora (Grundkiewicz and Junczys-Dowmunt, 2019; White and Rozovskaya, 2020). Conversely, MixEdit is proposed to augment high-quality realistic data and can be combined with other traditional data augmentation methods introduced in Section 2 due to their orthogonality.

## 4 Experiments on English GEC

### 4.1 Experimental Settings

**Datasets and evaluation.** We decompose the training of the baseline model into three stages following Zhang et al. (2022b). We train the model on 1) the CLang8 dataset (Rothe et al., 2021), 2) the FCE dataset (Yannakoudakis et al., 2011), the NUCLE dataset (Dahlmeier et al., 2013) and the W&I+LOCNESS train-set (Bryant et al., 2019). 3) We finally fine-tune the model on high-quality W&I-LOCNESS. As for the traditional data augmentation methods using extra corpora (i.e., DN, PN, BT and RT), we construct a pseudo dataset using the seed corpus Gigaword [4], which has been proven to be the best among three seed corpora by Kiyono et al. (2020). In total, we generate 8M pseudo data with the same target sentences for four data augmentation methods, which is used for the pre-training of GEC models.

For evaluation, we reports the results on the CoNLL-14 test set (Ng et al., 2014) evaluated by

---

[3] We leave more detailed analysis to Section 4.3.
[4] https://catalog.ldc.upenn.edu/LDC2011T07

| Dataset | #Sentences | Usage |
|---|---|---|
| Gigaword | 8,000,000 | Pre-training |
| CLang8 | 2,372,119 | Fine-tuning I |
| FCE | 34,490 | Fine-tuning II |
| NUCLE | 57,151 | Fine-tuning II |
| W&I+LOCNESS | 34,308 | Fine-tuning II&III |
| BEA-2019-*Dev* | 4,384 | Validation |
| BEA-2019-*Test* | 4,477 | Testing |
| CoNLL-2014-*Test* | 1,312 | Testing |

Table 2: Statistics of English GEC datasets. Gigaword is only available for DN, PN, BT and RT.

$\mathrm{M}^2$ scorer (Dahlmeier and Ng, 2012) and the BEA-19 test set (Bryant et al., 2019) evaluated by ER-RANT. The results are averaged over three runs with different random seeds, and the BEA-19 dev set serves as the validation set. We provide statistics of all the involved datasets in Table 2.

**GEC backbone model.** We adopt Transformer-based BART-Large (Lewis et al., 2020) as our backbone model, which has been shown as a strong baseline for GEC (Katsumata and Komachi, 2020; Zhang et al., 2022b). We acquire subwords by byte-pair-encoding (BPE) (Sennrich et al., 2016b) algorithm. We apply the Dropout-Src mechanism (Junczys-Dowmunt et al., 2018) to source-side word embeddings following Zhang et al. (2022b). All experiments are conducted using the Fairseq (Ott et al., 2019) public toolkit. Most of the hyperparameter settings are identical to Zhang et al. (2022b), which are provided in Appendix A.1.

**Data Augmentation.** We examine and analyze four mainstream GEC data augmentation methods discussed in Section 2, as well as our proposed MixEdit in Section 3.2. We introduce an extra pre-training stage that utilizes the pseudo datasets generated by each data augmentation method. Further details regarding the experimental settings for augmentation methods can be found in Appendix A.2.

### 4.2 Results of GEC Data Augmentation

Table 3 showcases the results of each data augmentation method. Based on the baseline model, PN achieves the highest $F_{0.5}$ score on CoNLL-14, while BT obtains the highest $F_{0.5}$ score on BEA-19. In contrast, DN produces lower $F_{0.5}$ score than the baseline in both test sets, which contradicts the findings of Kiyono et al. (2019) that DN can benefit the GEC performance of scratch transformer models. We attribute the phenomenon to the fact that

| | System | Extra Data Size | Transformer Layer, Hidden, FFN | CoNLL-14-*test* | | | BEA-19-*test* | | |
|---|---|---|---|---|---|---|---|---|---|
| | | | | P | R | $F_{0.5}$ | P | R | $F_{0.5}$ |
| w/o PLM | Kiyono et al. (2019)[°] | 70M | 12+12,1024,4096 | 67.9 | 44.1 | 61.3 | 65.5 | 59.4 | 64.2 |
| | Lichtarge et al. (2020)[△▲] | 340M | 12+12,1024,4096 | 69.4 | 43.9 | 62.1 | 67.6 | 62.5 | 66.5 |
| | Stahlberg and Kumar (2021)[△▲□] | 540M | 12+12,1024,4096 | 72.8 | 49.5 | **66.6** | 72.1 | 64.4 | **70.4** |
| w/ PLM | Kaneko et al. (2020)[°] | 70M | 12+12,1024,4096 | 69.2 | 45.6 | 62.6 | 67.1 | 60.1 | 65.6 |
| | Katsumata and Komachi (2020) | - | 12+12,1024,4096 | 69.3 | 45.0 | 62.6 | 68.3 | 57.1 | 65.6 |
| | Omelianchuk et al. (2020)[◇] | 9M | 12+0,768,3072 | 77.5 | 40.1 | 65.3 | 79.2 | 53.9 | 72.4 |
| | Rothe et al. (2021)[♡] | 2.4M | 12+12,1024,4096 | - | - | 66.1 | - | - | 72.1 |
| | Sun et al. (2021)[★] | 300M | 12+2,1024,4096 | 71.0 | 52.8 | 66.4 | - | - | 72.9 |
| | BART Baseline[♡] (Zhang et al., 2022b) | 2.4M | 12+12,1024,4096 | 73.6 | 48.6 | 66.7 | 74.0 | 64.9 | 72.0 |
| | SynGEC[♡] (Zhang et al., 2022b) | 2.4M | 12+12,1024,4096 | 74.7 | 49.0 | **67.6** | 75.1 | 65.5 | **72.9** |
| | **Our BART Baseline**[♡] | 2.4M | 12+12,1024,4096 | 73.8 | 47.6 | 66.5 | 74.4 | 63.7 | 72.0 |
| | +DN[♣♡] | 10.4M | 12+12,1024,4096 | 70.3 | 50.3 | 65.1 | 72.6 | 64.6 | 70.9 |
| | +PN[♣♡] | 10.4M | 12+12,1024,4096 | 73.3 | 51.1 | **67.4** | 75.0 | 65.1 | 72.8 |
| | +BT[♣♡] | 10.4M | 12+12,1024,4096 | 72.7 | 51.3 | 67.1 | 75.1 | 65.9 | **73.0** |
| | +RT[♣♡] | 10.4M | 12+12,1024,4096 | 73.0 | 48.7 | 66.4 | 75.1 | 64.0 | 72.5 |
| | **MixEdit**[♡] | 2.4M | 12+12,1024,4096 | 75.6 | 46.8 | $67.3^{\uparrow}$ | 76.4 | 62.7 | $73.2^{\uparrow}$ |
| | +DN[♣♡] | 10.4M | 12+12,1024,4096 | 72.6 | 48.3 | $66.0^{\uparrow}$ | 74.4 | 63.0 | $71.8^{\uparrow}$ |
| | +PN[♣♡] | 10.4M | 12+12,1024,4096 | 75.1 | 48.1 | $\mathbf{67.5}^{\uparrow}$ | 75.3 | 64.7 | $72.9^{\uparrow}$ |
| | +BT[♣♡] | 10.4M | 12+12,1024,4096 | 74.4 | 48.4 | $67.2^{\uparrow}$ | 76.4 | 63.7 | $\mathbf{73.4}^{\uparrow}$ |
| | +RT[♣♡] | 10.4M | 12+12,1024,4096 | 75.2 | 47.2 | $67.2^{\uparrow}$ | 75.9 | 63.3 | $72.9^{\uparrow}$ |

Table 3: The results of baselines and GEC data augmentation methods for **single-model**. **Layer**, **Hidden** and **FFN** denote the depth, hidden size and feed-forward network size of Transformer. "Our BART Baseline" is re-implemented from the open-source "BART Baseline", both of which are under the same experimental setting, making them fairly comparable. $^{\uparrow}$means the performance of "**w/ MixEdit**" is better than that of its "**w/o MixEdit**" counterpart. Besides the public human-annotated training data, private and/or pseudo data sources are also widely used in GEC systems, including: [°]BT pseudo data from Gigaword (70M sentences), [△]Wikipedia revision histories (170M), [▲]RT pseudo data from Wikipedia (170M), [□]BT pseudo data from Colossal Clean Crawled Corpus (200M), [◇]DN psuedo data from one-billion-word (9M), [★]BT and DN pseudo data (300M), [♡]cleaned version of Lang8 (2.4M), [♣]our psuedo data from Gigaword (8M).

the pre-training task of BART is similar to DN, and it is unnecessary and possibly harmful to pre-train BART again with task-independent noise. Finally, RT performs slightly better on BEA-19 but slightly worse on CoNLL-14.

The bottom group lists the results of MixEdit and its combination with each traditional GEC data augmentation method. Without extra monolingual corpora, our MixEdit achieves 67.3/73.2 $F_{0.5}$ scores, which are on par with or marginally superior to traditional methods using the same backbone model. Notably, MixEdit also complements these traditional methods. Each augmentation method based on MixEdit produces higher $F_{0.5}$ scores than its counterpart, with PN and BT **w/ MixEdit** achieving the highest $F_{0.5}$ scores on CoNLL-14 (67.5) and BEA-19 (73.4), respectively.

## 4.3 Analysis

The mechanism through which these corruptions work remains unclear, although the effectiveness of GEC data augmentation is well-established. In this section, we investigate the relationship between data augmentation and performance through the lens of quantified Affinity and Diversity measures, seeking to gain insight into the mechanisms underlying GEC data augmentation.

To ensure a fair comparison, we avoid introducing extra monolingual corpora that may influence result due to nuisance variables such as text domain. Instead, we apply data augmentation to GEC realistic datasets comprising FCE, NUCLE and W&I+LOCNESS (collectively referred to as BEA-train). Specifically, we retain the target sentences from BEA-train, and generate pseudo source sentences by enforcing data augmentation strategies, resulting in a set of pseudo datasets with identical targets but different sources. It is worth noting that we remove the dynamic training and the regularization of MixEdit to facilitate a fair comparison with other data augmentation methods. We respectively train GEC models on these pseudo datasets and report the results in Table 4.

**Affinity is positively correlated with the performance.** At a high level, Affinity of MixEdit, PN and BT, which achieve the highest $F_{0.5}$ scores, is much higher than that of DN and RT. We compute the Pearson's correlation coefficient for the data

| Method | Affinity↑ | Diversity | P | R | $F_{0.5}$ |
|---|---|---|---|---|---|
| Baseline | - | 8.78 | **56.03** | 37.60 | **51.03** |
| DN | 0.41 | 10.70 | 20.49 | 12.94 | 18.35 |
| RT | 0.75 | 10.54 | 20.59 | 38.41 | 22.70 |
| PN | | | | | |
| Round=1 | 1.91 | 7.46 | 35.46 | 16.35 | 28.74 |
| Round=2 | **1.93** | 7.56 | **39.43** | 23.71 | 34.82 |
| Round=4 | 1.90 | 7.70 | 39.41 | 29.41 | 36.90 |
| Round=8 | 1.77 | 7.92 | 42.14 | 37.51 | **41.13** |
| Round=16 | 1.59 | 8.24 | 33.83 | **39.24** | 34.79 |
| BT | | | | | |
| $\beta_{random}$=0 | 0.65 | 6.96 | 22.20 | 7.36 | 15.82 |
| $\beta_{random}$=3 | 1.47 | 7.67 | 44.11 | 32.46 | 41.16 |
| $\beta_{random}$=6 | **1.57** | 8.22 | **49.29** | 42.99 | **47.89** |
| $\beta_{random}$=9 | 1.53 | 8.59 | 44.69 | 47.76 | 45.27 |
| $\beta_{random}$=12 | 1.45 | **8.84** | 41.43 | **50.64** | 43.00 |
| MixEdit (Ours) | **2.33** | 8.52 | **57.72** | 33.24 | **50.31** |

Table 4: Affinity and Diversity of data augmentation methods. The baseline model is trained using realistic BEA-train dataset.

augmentation methods involved: DN, PN, BT, RT and MixEdit. For PN and BT, we choose the hyper-parameter configurations that yield the highest F0.5 scores. The Pearson's correlation coefficient between $F_{0.5}$ and Affinity is 0.9485. The results indicate a strong correlation between Affinity and $F_{0.5}$ on BEA-train. It should be noted that, despite having lower Affinity, BT achieves a higher $F_{0.5}$ score than PN. We attribute this to the advantage of BT in learning the distribution of grammatical errors using Seq2Seq models instead of adding errors crudely. MixEdit also skillfully avoids the drawbacks of PN by strategically applying label-preserving perturbations, resulting in an approximate $F_{0.5}$ score with the baseline.

**Diversity is responsible for the trade-off between Precision and Recall.** We qualitatively investigate the effect of Diversity on GEC performance within a fixed method. The trade-off of Diversity is apparent when adjusting certain hyper-parameters responsible for Diversity (Round for PN and $\beta$ for BT). As Diversity increases, Precision and $F_{0.5}$ initially increase and then decrease, reaching their peak at an appropriate setting. Meanwhile, Recall continues to increase. For example, PN and BT reach their peak $F_{0.5}$ score at *Round=8* and $\beta_{random}$=6, respectively, falling at an intermediate value of Diversity.

Therefore, we argue that an excellent data augmentation technique should have high Affinity and appropriate Diversity, which motivated our proposed MixEdit. The high Affinity of MixEdit stems from the fact that the distribution of pseudo grammatical errors it generates is the same as that of the

| | CoNLL-14-*test* | BEA-19-*test* |
|---|---|---|
| | P/R/$F_{0.5}$ | P/R/$F_{0.5}$ |
| MixEdit | 76.81/45.00/67.30 | 76.37/62.71/**73.18** |
| w/o Error Pattern | 73.08/48.46/66.35 | 75.22/63.08/72.44 |
| w/o Consistency Loss | 74.73/47.80/67.16 | 76.04/63.39/73.12 |
| w/o Dynamic Generation | 75.41/46.74/67.17 | 75.73/62.35/72.61 |
| w Pattern Noise | 74.70/49.73/**67.88** | 74.78/64.67/72.51 |

Table 5: Ablation results of MixEdit.

| $\alpha$ | P | R | $F_{0.5}$ |
|---|---|---|---|
| **0.5** | 55.96 | 41.35 | 52.27 |
| **0.8** | 55.65 | 41.35 | 52.05 |
| **1.0** | 57.96 | 39.00 | **52.83** |
| **1.2** | 56.75 | 40.93 | 52.68 |
| **2.0** | 55.88 | 41.30 | 52.20 |

Table 6: Results of various $\alpha$ on BEA-train.

| $\beta$ | P | R | $F_{0.5}$ |
|---|---|---|---|
| **0.5** | 57.56 | 38.51 | 52.38 |
| **1.0** | 57.96 | 39.00 | **52.83** |
| **2.0** | 56.69 | 40.59 | 52.52 |

Table 7: Results of various $\beta$ on BEA-train.

ground truth. MixEdit only replaces the original grammatical errors with augmented ones, maintaining the same error density in the augmented dataset. As a result, the diversity of the augmented dataset closely resembles that of the original dataset.

To further explore the relationship between Affinity/Diversity and performance across different datasets, we also conduct additional experiments on English CLang8 and Chinese HSK. The relationship between Affinity/Diversity and performance is similar to the results on BEA-train, which are provided in Appendix B.1. Additionally, we provide further analysis on the complementary effectiveness of pseudo data in Appendix B.2.

### 4.4 Ablation Study

**Decomposition of MixEdit.** We explore the effectiveness of each component of our proposed MixEdit by conducting ablation studies shown in Table 5. Specifically, for the "w/o Error Pattern" variant, we randomly mask tokens in the sentence instead of sampling grammatical errors from the Error Pattern Pool. For "w/o Consistency Loss", we remove the consistency loss. For "w/o Dynamic Generation", we always generate fixed pseudo data for fine-tuning GEC models. Additionally, we attempt to incorporate PN into generating dynamic pseudo data in the fine-tuning stage. The results demonstrate that task-specific information of error pattern is important to constructing high-quality

| System | Extra Data Size | NLPCC-18-*test* | | | MuCGEC-*test* | | |
|---|---|---|---|---|---|---|---|
| | | P | R | $F_{0.5}$ | P | R | $F_{0.5}$ |
| **Zhang et al. (2022a)** | - | 42.88 | 30.19 | 39.55 | 43.81 | 28.56 | 39.58 |
| **SynGEC** (Zhang et al., 2022b) | - | 49.96 | 33.04 | 45.32 | 54.69 | 29.10 | 46.51 |
| **Our Baseline** | - | 49.81 | 31.57 | 44.65 | 54.24 | 29.67 | 46.53 |
| **+DN** | 8M | 49.57 | 31.80 | 44.59 | 54.93 | 29.61 | 46.91 |
| **+PN** | 8M | 50.15 | 35.27 | **46.25** | 55.83 | 30.15 | **47.71** |
| **+BT** | 8M | 47.64 | 36.43 | 44.88 | 54.82 | 30.27 | 47.17 |
| **+RT** | 8M | 51.06 | 30.74 | 45.09 | 56.96 | 27.41 | 46.86 |
| **MixEdit** | - | 49.58 | 32.93 | $45.03^{\uparrow}$ | 55.25 | 29.30 | $46.94^{\uparrow}$ |
| **+DN** | 8M | 50.46 | 30.55 | $44.64^{\uparrow}$ | 56.48 | 28.12 | $47.00^{\uparrow}$ |
| **+PN** | 8M | 52.26 | 33.37 | $\mathbf{46.94}^{\uparrow}$ | 56.99 | 29.73 | $\mathbf{48.16}^{\uparrow}$ |
| **+BT** | 8M | 48.99 | 35.45 | $45.52^{\uparrow}$ | 54.72 | 31.76 | $47.81^{\uparrow}$ |
| **+RT** | 8M | 51.19 | 34.94 | $45.68^{\uparrow}$ | 55.04 | 30.53 | $47.43^{\uparrow}$ |

Table 8: **Single-model** results on Chinese datasets. All models are initialized with pre-trained Chinese BART weights. $^{\uparrow}$ means the performance of "**w/ MixEdit**" is better than its "**w/o MixEdit**" counterpart.

pseudo data. Dynamic Generating is another critical factor for the success of MixEdit, since it can improve the diversity of pseudo data without the loss of Affinity. Surprisingly, adding pattern noise improves $F_{0.5}$ on CoNLL-14-*test*, with a performance reduction on BEA-19-*test*. Given that CoNLL-14 is centered around essays written by language learners, our speculation is that it includes a greater number of stereotypical grammatical errors. As a result, incorporating pseudo data with a higher corruption ratio can improve performance on CoNLL-14. We investigate how varying PN corruption ratios affect the performance of the GEC model in Appendix B.3.

**Sensitivity to hyperparameters.** We investigate the sensitivity to the choice of the hyperparameters $\alpha$ and $\beta$ introduced in Equation 6. We explore the optimal values of them on BEA-train. The results of various $\alpha$ and $\beta$ are reported in Table 6 and Table 7, respectively, where the optimal hyperparameter setting is $\alpha = 1.0$ and $\beta = 1.0$.

## 5 Experiments on Chinese GEC

### 5.1 Experimental Settings

**Datasets and evaluation.** We adopt the seed corpus *news2016zh* [5] to generate 8M pseudo data for each traditional data augmentation method, similar to our English experiments. Following (Zhang et al., 2022a), we fine-tune GEC models on the Chinese Lang8 (Zhao et al., 2018) and HSK (Zhang, 2009) datasets. We reports the results on NLPCC-2018-*test* (Zhao et al., 2018) evaluated by $M^2$ scorer, and MuCGCE-*test* (Zhang et al., 2022a)

[5] https://github.com/brightmart/nlp_chinese_corpus

| Dataset | #Sentences | Usage |
|---|---|---|
| **News2016zh** | 8,000,000 | Pre-training |
| **Lang8** | 1,220,906 | Fine-tuning |
| **HSK** | 156,870 | Fine-tuning |
| **MuCGEC-*dev*** | 1,125 | Validation |
| **MuCGEC-*test*** | 5,938 | Testing |
| **NLPCC-18-*test*** | 2,000 | Testing |

Table 9: Statistics of Chinese GEC datasets.

evaluated by ChERRANT. Table 9 provides statistics for the aforementioned datasets.

**GEC backbone model.** We employ Chinese BART (Shao et al., 2021) as our backbone model [6]. We retained the original vocabulary because the updated version of Chinese BART has already incorporated a larger vocabulary, and therefore no modifications were necessary.

### 5.2 Results

Table 8 presents our results. With the exception of PN, all data augmentation methods improve the $F_{0.5}$ score, demonstrating that low-affinity data augmentation methods can negatively impact the performance of pre-trained models like BART. Moreover, incorporating MixEdit with traditional methods can further increase the $F_{0.5}$ scores on both evaluation datasets. These findings suggest that MixEdit is a general technique that can be effectively employed in various languages and combined with other data augmentation methods.

## 6 Related Works

**Data Augmentation.** Data augmentation encompasses methods of increasing training data diver-

[6] https://huggingface.co/fnlp/bart-large-chinese

sity without requiring additional data collection. It has become a staple component in many downstream NLP tasks. Most researches apply a range of perturbation techniques to increase training scale, with the aim of reducing overfitting and improving the generalization of models (Feng et al., 2021; Shorten and Khoshgoftaar, 2019). Furthermore, many researchers develop various data augmentation strategies based on unverified heuristics. Despite the success of data augmentation, there appears to be a lack of research on why it works. To explore the effectiveness of these augmentation heuristics, Kashefi and Hwa (2020) proposed to quantify an augmentation heuristic for text classification based on the idea that a good heuristic should generate "hard to distinguish" samples for different classes. Gontijo-Lopes et al. (2020) introduced interpretable and easy-to-compute measures to qualify an augmentation heuristic in the field of Computer Vision (Liu et al., 2022).

**Pseudo Data Generation.** Increasing the scale of training data has been instrumental in improving GEC systems. However, the lack of high-quality publicly available data remains a challenge in low-resource settings (Ma et al., 2022; Li et al., 2022a). To mitigate this issue, recent studies focus on generating pseudo data from clean monolingual corpora. A common approach is to artificially perturb a grammatically correct sentence through random word or character-level insertion, substitution, or deletion (Kiyono et al., 2020; Xu et al., 2019; Zhou et al., 2020), using spell checkers (Grundkiewicz et al., 2019) or error patterns extracted from realistic datasets (Choe et al., 2019). However, these rule-based methods struggle to emulate human-made errors, which can lead to low sample efficiency problem and performance degradation (Yuan and Felice, 2013). Recent studies utilize models to generate genuine pseudo data, such as backtranslation (Xie et al., 2018) and round-trip translation (Zhou et al., 2020). On the other hand, Koyama et al. (2021a) shown that the performance of GEC models improves when pseudo datasets generated by various backtranslation models are combined. Stahlberg and Kumar (2021) proposed to generate pseudo data based on a given error type tag using the Seq2Edit model (Stahlberg and Kumar, 2020). Different from all these studies, we focus on generating high-quality pseudo data without extra corpora.

## 7 Conclusion

This paper introduces two interpretable and computationally efficient measures, Affinity and Diversity, to investigate how data augmentation improves GEC performance. Our findings demonstrate that an excellent GEC data augmentation strategy characterized by high Affinity and appropriate Diversity can better improve the performance of GEC models. Inspired by this, we propose MixEdit, which does not require extra monolingual corpora. Experiments on mainstream datasets in two languages show that MixEdit is effective and complementary to traditional data augmentation methods.

## Limitations

One shall cautiously consider that our proposed measures, Affinity and Diversity, are a tool for gaining a new perspective on understanding GEC data augmentation. Though positive correlations are observed between Affinity and performance, it should not be relied upon as a precise predictor for comparing data augmentation methods. Additionally, it is worth noting that our proposed MixEdit approach is only applicable to realistic datasets, where it can generate label-preserving grammatical errors. Despite these limitations, we believe that our findings provide a solid foundation for further scientific investigation into GEC data augmentation.

## Ethics Statement

In this paper, we revisit the effectiveness of traditional data augmentation methods for GEC, including direct noise (DN), pattern noise (PN), backtranslation (BT) and round-trip translation (RT). The source data for these methods is obtained exclusively from publicly available project resources on legitimate websites, without any involvement of sensitive information. Additionally, all the baselines and datasets used in our experiments are also publicly accessible, and we have acknowledged the corresponding authors by citing their work.

## Acknowledgements

This research is supported by National Natural Science Foundation of China (Grant No.62276154), Research Center for Computer Network (Shenzhen) Ministry of Education, the Natural Science Foundation of Guangdong Province (Grant No. 2023A1515012914), Basic Research Fund of Shenzhen City (Grant No. JCYJ20210324120012033

and JSGG20210802154402007), the Major Key Project of PCL for Experiments and Applications (PCL2021A06), and Overseas Cooperation Research Fund of Tsinghua Shenzhen International Graduate School (HW2021008).

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

# A Implementation Details

## A.1 Hyper-parameters

| Configuration | Value |
|---|---|
| **Pre-training** | |
| Backbone | BART-large (Lewis et al., 2020) |
| Devices | 4 Tesla A100 GPU (80GB) |
| Epochs | 60 |
| Batch size per GPU | 4096 tokens |
| Optimizer | Adam (Kingma and Ba, 2014) $(\beta_1 = 0.9, \beta_2 = 0.999, \epsilon = 1 \times 10^{-8})$ |
| Learning rate | $3 \times 10^{-5}$ |
| Warmup updates | 2000 |
| Max source length | 1024 |
| Dropout | 0.3 (English); 0.2 (Chinese) |
| Dropout-src | 0.2 |
| **Fine-tuning** | |
| Weights of Loss | $\alpha$=1.0, $\beta$=1.0 |
| Learning rate | $3 \times 10^{-5}, 5 \times 10^{-6}, 3 \times 10^{-6}$ (English) $3 \times 10^{-5}$ (Chinese) |
| Warmup updates | 2000 |
| **Inference** | |
| Beam size | 12 |

Table 10: Hyper-parameter values used in our experiments.

We list the main hyper-parameters in Table 10. For the pre-training stage, we follow the same hyper-parameters as described in (Zhang et al., 2022b). We introduce MixEdit throughout the three-stage fine-tuning. To determine the optimal balancing weights for the training objective, we experiment with various values for $\alpha$ within {0.5, 1.0, 1.2, 2.0} and $\beta$ within {0.5, 1.0, 2.0}. Our experiments reveal that the configuration with $\alpha = 1.0$ and $\beta = 1.0$ achieves the highest $F_{0.5}$ score on BEA-19 and CoNLL-14, which is used as the default settings in all our experiments.

## A.2 Details of GEC Data Augmentation

We follow the default experimental settings of GEC data augmentation methods as proposed in their papers or source code. In our experiments, we set $\mu_{\text{mask}} = 0.3$ for DN, which has been proven to be the best by Kiyono et al. (2020). For PN, we generate pseudo data by following the default instructions provided in the open-source project [7]. We train the BT model, which is initialized with the weights of BART-large, on the CLang8 dataset, setting $\beta_{\text{random}} = 6$ to yields the best performance as evidenced in Section 4.3. As for RT, we generate pseudo data via Chinese as the bridge language, leveraging two off-the-shelf en-zh [8] and zh-

---

[7] https://github.com/kakaobrain/helo-word
[8] https://huggingface.co/Helsinki-NLP/opus-mt-en-zh

en [9] translation models (Tiedemann and Thottingal, 2020).

# B Extra Experiments

## B.1 Affinity and Diversity of Extra Datasets

| Method | Affinity$^\uparrow$ | Diversity | P | R | F$_{0.5}$ |
|---|---|---|---|---|---|
| **Baseline** | - | 8.81 | **63.04** | 44.69 | **58.25** |
| **DN** | 0.33 | 11.31 | 28.52 | 24.48 | 27.60 |
| **RT** | 0.53 | 11.14 | 18.43 | 37.16 | 20.50 |
| **PN** | | | | | |
| **Round=1** | **3.74** | 7.63 | **62.66** | 28.32 | 50.43 |
| **Round=2** | 3.55 | 7.79 | 60.24 | 34.12 | **52.24** |
| **Round=4** | 3.17 | 8.05 | 56.70 | 38.58 | 51.83 |
| **Round=8** | 2.65 | 8.47 | 49.70 | 43.36 | 48.29 |
| **Round=16** | 1.92 | **9.10** | 40.59 | **48.21** | 41.92 |
| **BT** | | | | | |
| $\beta_{\mathrm{random}}$=0 | 1.04 | 7.04 | 57.50 | 13.96 | 35.41 |
| $\beta_{\mathrm{random}}$=3 | 3.59 | 7.60 | **62.63** | 38.94 | **55.83** |
| $\beta_{\mathrm{random}}$=6 | **3.73** | 8.15 | 57.73 | 46.71 | 55.13 |
| $\beta_{\mathrm{random}}$=9 | 3.08 | 8.55 | 53.07 | 51.19 | 52.68 |
| $\beta_{\mathrm{random}}$=12 | 2.55 | **8.87** | 49.38 | **52.63** | 50.00 |
| **MixEdit** (Ours) | **19.29** | 8.85 | **64.50** | 38.32 | 56.75 |

Table 11: Affinity and Diversity of data augmentation methods. All the models are trained using realistic or pseudo English CLang8 datasets.

| Method | Affinity$^\uparrow$ | Diversity | P | R | F$_{0.5}$ |
|---|---|---|---|---|---|
| **Baseline** | - | 8.78 | **49.10** | 25.30 | **41.33** |
| **DN** | 0.39 | 10.10 | 28.84 | 26.56 | 28.35 |
| **RT** | 0.46 | 10.57 | 19.39 | 29.84 | 20.84 |
| **PN** | | | | | |
| **Round=1** | **3.07** | 6.40 | **45.84** | 08.86 | 24.98 |
| **Round=2** | 2.73 | 6.59 | 43.54 | 13.28 | 29.91 |
| **Round=4** | 2.25 | 6.91 | 44.96 | 15.12 | 32.24 |
| **Round=8** | 1.43 | 7.43 | 42.83 | 16.89 | 32.76 |
| **Round=16** | 1.17 | **8.20** | 39.82 | **21.24** | 33.89 |
| **BT** | | | | | |
| $\beta_{\mathrm{random}}$=0 | 1.30 | 5.44 | 27.07 | 03.34 | 11.18 |
| $\beta_{\mathrm{random}}$=3 | **3.00** | 7.24 | **45.48** | 15.78 | 33.04 |
| $\beta_{\mathrm{random}}$=6 | 2.73 | 8.10 | 41.72 | 26.32 | **37.35** |
| $\beta_{\mathrm{random}}$=9 | 2.17 | 8.83 | 37.83 | 30.27 | 36.03 |
| $\beta_{\mathrm{random}}$=12 | 1.75 | **9.45** | 32.27 | **32.59** | 32.33 |
| **MixEdit** (Ours) | **3.95** | 8.52 | **47.81** | 18.31 | 36.15 |

Table 12: Affinity and Diversity of data augmentation methods. All the models are trained using realistic or pseudo Chinese HSK datasets.

We conduct extra experiments illustrated in Section 4.3 on English CLang8 and Chinese HSK datasets. The results are listed in Table 11 and Table 12, respectively. Surprisingly, backtranslation (BT) achieves the highest F0.5 on Chinese HSK. We speculate that Chinese grammatical errors are inherently more intricate, providing an advantage for BT as it can generate pseudo grammatical errors that are closer to authentic ones. Nonetheless, it is

---

| Method | Data | Aff.$^\uparrow$ | Div. | P | R | F$_{0.5}$ |
|---|---|---|---|---|---|---|
| **Baseline** | 124K | - | 8.78 | 56.03 | 37.60 | 51.03 |
| **DN** | 249K | 0.41 | 10.70 | 53.89 | 39.72 | 50.30 |
| **PN** | | | | | | |
| **RT** | 249K | 0.75 | 10.54 | 50.32 | 39.82 | 47.80 |
| **Round=1** | 249K | 1.91 | 7.46 | 56.64 | 36.75 | 51.11 |
| **Round=2** | 249K | **1.93** | 7.56 | 56.07 | 40.15 | **51.95** |
| **Round=4** | 249K | 1.90 | 7.70 | 53.88 | 43.04 | 51.30 |
| **Round=8** | 249K | 1.77 | 7.92 | 54.03 | 41.32 | 50.90 |
| **Round=16** | 249K | 1.59 | 8.24 | 54.43 | 40.88 | 51.04 |
| **BT** | | | | | | |
| $\beta_{\mathrm{random}}$=0 | 249K | 0.65 | 6.96 | 56.46 | 33.62 | 49.71 |
| $\beta_{\mathrm{random}}$=3 | 249K | 1.47 | 7.67 | 54.94 | 42.87 | 52.01 |
| $\beta_{\mathrm{random}}$=6 | 249K | **1.57** | 8.22 | 54.04 | 46.38 | **52.31** |
| $\beta_{\mathrm{random}}$=9 | 249K | 1.53 | 8.59 | 53.63 | 45.76 | 51.84 |
| $\beta_{\mathrm{random}}$=12 | 249K | 1.45 | 8.84 | 53.79 | 43.03 | 51.22 |
| **MixEdit (Static)** | 249K | **2.33** | 8.52 | 57.98 | 37.78 | **52.38** |
| **MixEdit (Dynamic)** | - | - | - | 57.96 | 39.00 | **52.83** |

Table 13: Results on the combination of the realistic and pseudo BEA-train datasets for various data augmentation methods. **MixEdit (Dynamic)** generate pseudo data on-the-fly during fine-tuning.

worth noting that BT relies on an additional model to generate grammatical errors, which introduces efficiency concerns. The Pearson's correlation coefficients between F0.5 and Affinity are 0.6239 on CLang8, and 0.7717 on HSK. The results indicate a strong or moderate correlation between Affinity and F0.5 on different datasets, demonstrating the generalization of our proposed approach.

## B.2 Complementary Effectiveness of Pseudo Data

We have analyzed the effectiveness of GEC data augmentation methods from the lens of Affinity and Diversity measures in Section 4.3. However, one may argue that MixEdit gains unfair advantages since the information about the positions of grammatical errors is only visible to MixEdit. To this end, we investigate the complementary effectiveness of pseudo data in this section. Specifically, we construct a combination dataset by appending the realistic data of BEA-train to the pseudo data generated by data augmentation methods, where a target sentence correspond to two source sentences.

We train GEC models using the combination dataset and report the results in Table 13. Similarly, DN and RT perform worse than the baseline since these methods inject considerable undesired noise, which makes the model prone to inaccurate corrections. PN arrives its peek of F$_{0.5}$ score at Round=2, rather than Round=8 in Table 4. BT arrives the peek at $\beta_{\mathrm{random}} = 6$, and most selections of $\beta_{\mathrm{random}}$ can improve GEC models. MixEdit (Static) performs the best among all static data augmentation meth-

---

[9] https://huggingface.co/Helsinki-NLP/opus-mt-zh-en

| Corruption Ratio | CoNLL-14-*test* P/R/F$_{0.5}$ | BEA-19-*test* P/R/F$_{0.5}$ |
|---|---|---|
| 0.00 | 76.81/45.00/67.30 | 76.37/62.71/**73.18** |
| 0.02 | 74.94/48.70/67.65 | 75.14/64.04/72.62 |
| 0.05 | 74.70/49.73/**67.88** | 74.78/64.67/72.51 |
| 0.10 | 72.42/50.68/66.70 | 73.70/65.25/71.84 |
| 0.15 | 73.26/48.95/66.64 | 72.96/64.46/71.08 |

Table 14: Results of incorporating PN into MixEdit. The Corruption Ratio indicates the probability of adding grammatical errors to a token in the source sentence.

ods. Furthermore, MixEdit (Dynamic) achieves the highest F$_{0.5}$ score, demonstrating the effectiveness of dynamic pseudo data construction.

### B.3 Incorporating PN into MixEdit

We also explore the effectiveness of incorporating PN into MixEdit in a dynamic manner. In this approach, we applying PN after running MixEdit. This means that we randomly add grammatical errors to the pseudo source generated by MixEdit. We report the results of varying PN corruption ratios in Table 14. Our findings suggest that when the correction ratio is low, PN can benefit GEC models on CoNLL-14-*test*. However, PN decreases the F$_{0.5}$ scores on BEA-19-*test* regardless of different correction ratios. We attribute this to the different characteristics of the two evaluation datasets, where CoNLL-14 contains more typical grammatical errors written by language learners.