# OpenReview forum: "MixEdit: Revisiting Data Augmentation and Beyond for Grammatical Error Correction"
_EMNLP/2023/Conference — EMNLP 2023 Findings_

### Official Review · Reviewer_RBnA · 2023-07-24

**Soundness:** 3

**Excitement:**

3: Ambivalent: It has merits (e.g., it reports state-of-the-art results, the idea is nice), but there are key weaknesses (e.g., it describes incremental work), and it can significantly benefit from another round of revision. However, I won't object to accepting it if my co-reviewers champion it.

**Paper Topic And Main Contributions:**

This paper presents two metrics that characterizes pseudo data for grammatical error correction (GEC), namely, affinity and diversity.
Base on these two metrics, this paper proposes a novel method for data augmentation (MixEdit).
MixEdit applies data augmentation to real training data by generating label-preserving grammatical errors.
The contribution of this paper is that MixEdit achieves the state-of-the-art result on three benchmark datasets.

**Questions For The Authors:**

- (1) Table 8: I wonder if there exists a V100 card with 80GB memory. Is this a typo?
- (2) Is MixEdit sensitive to the choice of hyperparameter alpha and beta? I wish to see the performance plot.
- (3) Depending on the error pattern, the length of original sequence and edited sequence may not match. In such case, how do you compute P_avg?
- (4) Why do the authors call their method MixEdit? To what extent does it "Mix" and "Edit"?

**Reasons To Accept:**

- The authors conduct experiments on multiple languages (English and Chinese). These experiments strongly support the effectiveness of the proposed method (MixEdit).
- The proposed method achieves the state-of-the-art performance on benchmark datasets.

**Reasons To Reject:**

## Diversity Metric?

My biggest concern with this paper is that the usefulness of the proposed metrics are questionable. On one hand, the analysis on Table 4 presents that high affinity value seems to consistently improve the model performance. On the other hand, there is no consistent trend for diversity metric. For example, high diversity does not seem to always improve the performance. Given this result, the authors recommends that an "appropriate" diversity is important. However, I believe that such recommendation is unfriendly for those who wishes to utilize these metrics; one still needs to conduct an extensive hyperparameter search on diversity to maximize the performance.

## Clarity Issue

I had difficulty with understanding the details of this paper.

(1): Notation is inconsistent. In Section 2, "x" and "y" represent ungrammatical source sentence and grammatical target sentence, respectively. However, in Section 3.1, "x" and "y" seem to represent different concepts. I assume that they represent a pair of corrections (L225). In addition, there is yet another notation in Figure 1. Here, "X" and "Y" are used for representing source and reference. As a result I was unable to understand how the authors compute both affinity and diversity.

(2): How does MixEdit manage to achieve diversity and affinity? I was unable to find a discussion on this aspect.

**Reproducibility:**

2: Would be hard pressed to reproduce the results. The contribution depends on data that are simply not available outside the author's institution or consortium; not enough details are provided.

**Reviewer Confidence:**

3: Pretty sure, but there's a chance I missed something. Although I have a good feel for this area in general, I did not carefully check the paper's details, e.g., the math, experimental design, or novelty.

**Typos Grammar Style And Presentation Improvements:**

- The authors might want to use a consistent notation for x and y throughout the paper
- Table 4 should be presented as a graph, to better illustrate the effectiveness of two proposed metrics.
- L344: "We acquire subwords by byte-pair-encoding": I am confused. Do the authors build the tokenizer by themselves, or do they use off-the-shelf tokenizer provided by BART model? I believe that the latter is more common practice.

---

> ### Author Rebuttal · Authors · 2023-08-28
>
> Thanks for your valuable comments. We attach great importance to your questions and hope to address your concerns.
>
> ---
>
> **Q1**: There is no consistent trend for diversity metric. It seems elusive to determine an appropriate diversity.
>
> **A1**: We believe that the optimal diversity of augmented datasets varies across evaluation datasets. For **L2-speaker** evaluation datasets with a higher density of grammatical errors, augmented datasets with greater diversity may be preferred, while for **native-speaker** evaluation datasets with lower error density, the **opposite** may be true. Therefore, determining the ideal level of diversity without conducting experiments proves to be challenging. Additionally, we would like to emphasize that our proposed Affinity/Diversity metrics aim to **offer a fresh perspective** on characterizing and comprehending GEC data augmentation, as mentioned in line 256.
> To support our claims regarding the relationship between Affinity/Diversity and GEC performance, we have **conducted additional experiments on English CLang8 and Chinese HSK**.
>
> **Table 1**: Affinity and Diversity of data augmentation methods on **English CLang8**. The experimental settings are identical to those mentioned in Section 4.3 of our paper, with the only difference being the choice of datasets.
>
> | Method        | Affinity | Diversity | P     | R     | F0.5  |
> | ------------- | -------- | --------- | ----- | ----- | ----- |
> | Baseline      | -        | 8.81      | 63.04 | 44.69 | 58.25 |
> | DN            | 0.33     | **11.31**     | 28.52 | 24.48 | 27.60 |
> | PN Round=1    | 3.74     | 7.63      | 62.66 | 28.32 | 50.43 |
> | PN Round=2    | 3.55     | 7.63      | 60.24 | 34.12 | 52.24 |
> | PN Round=4    | 3.17     | 8.05      | 56.70 | 38.58 | 51.83 |
> | PN Round=8    | 2.65     | 8.47      | 49.70 | 43.36 | 48.29 |
> | PN Round=16   | 1.92     | 9.10      | 40.59 | 48.21 | 41.92 |
> | BT $\beta=0$  | 1.04     | 7.04      | 57.50 | 13.96 | 35.41 |
> | BT $\beta=3$  | 3.59     | 7.60      | 62.63 | 38.94 | 55.83 |
> | BT $\beta=6$  | 3.73     | 8.15      | 57.73 | 46.71 | 55.13 |
> | BT $\beta=9$  | 3.08     | 8.55      | 53.07 | 51.19 | 52.68 |
> | BT $\beta=12$ | 2.55     | 8.87      | 49.38 | **52.63** | 50.00 |
> | RT            | 0.53     | 11.14     | 18.43 | 37.16 | 20.50 |
> | MixEdit       | **19.29**    | 8.85      | **64.50** | 38.32 | **56.75** |
>
> **Table 2**: Affinity and Diversity of data augmentation methods on **Chinese HSK**.
>
> | Method        | Affinity | Diversity | P     | R     | F0.5  |
> | ------------- | -------- | --------- | ----- | ----- | ----- |
> | Baseline      | -        | 8.78      | 49.10 | 25.30 | 41.33 |
> | DN            | 0.39     | 10.10     | 28.84 | 26.56 | 28.35 |
> | PN Round=1    | 3.07     | 6.40      | 45.84 | 8.86  | 24.98 |
> | PN Round=2    | 2.73     | 6.59      | 43.54 | 13.28 | 19.91 |
> | PN Round=4    | 2.25     | 6.91      | 44.96 | 15.12 | 32.24 |
> | PN Round=8    | 1.43     | 7.43      | 42.83 | 16.89 | 32.76 |
> | PN Round=16   | 1.17     | 8.20      | 39.82 | 21.24 | 33.89 |
> | BT $\beta=0$  | 1.30     | 5.44      | 27.07 | 3.34  | 11.18 |
> | BT $\beta=3$  | 3.00     | 8.24      | 45.48 | 15.78 | 33.04 |
> | BT $\beta=6$  | 2.73     | 8.10      | 41.72 | 26.32 | **37.35** |
> | BT $\beta=9$  | 2.17     | 8.8.3     | 37.83 | 30.27 | 36.03 |
> | BT $\beta=12$ | 1.75     | 9.45      | 32.27 | **32.59** | 32.33 |
> | RT            | 0.46     | **10.57**     | 19.39 | 29.84 | 20.84 |
> | MixEdit       | **3.95**     | 8.52      | **47.81** | 18.31 | 36.15 |
>
>
> The relationship between Affinity/Diversity and performance is **similar to the results on BEA-train shown in our paper**.
>
> ---
>
> **Q2**: Inconsistent Notation.
>
> **A2**: The symbols $\rm{x}$ and $\rm{y}$ in Section 2 represent source and target sentences, whereas $\boldsymbol{x}$ and $\boldsymbol{y}$ in Section 3.1 indicate an edit from $\boldsymbol{x}$ to $\boldsymbol{y}$. Additionally, the symbols $X, Y$ in Figure 1 should be **lowercase** to represent source and target sentences. The typo will be corrected in the final version.
>
> ---
>
> **Q3**: How does MixEdit manage to achieve diversity and affinity?
>
> **A3**: The explanation for MixEdit's high Affinity and appropriate Diversity is provided in Section 3.2, specifically in Lines 260-276. The high Affinity of MixEdit stems from the fact that the **distribution** of pseudo grammatical errors it generates is the **same** as that of the ground truth. MixEdit only replaces the original grammatical errors with augmented ones, **maintaining the same error density** in the augmented dataset. As a result, the diversity of the augmented dataset **closely resembles** that of the original dataset.
>
> ---
>
> **Q4**: Table 8: I wonder if there exists a V100 card with 80GB memory. Is this a typo?
>
> **A4**: It is a typo, and "V100" is actually "A100".
>
> ---
>
> **Q5**: Is MixEdit sensitive to the choice of hyperparameter alpha and beta?
>
> **A5**: In Appendix A.1, we explore the optimal values of $\alpha$ and $\beta$, which are chosen from the set {0.5 0.8, 1.0, 1.2, 2.0} and {0.5, 1.0, 2.0}, respectively. We conduct additional experiments on BEA-train and report the results below.
>
>
> **Table 3**: Results of various $\alpha$ on BEA-train.
>
> | $\alpha$ | P     | R     | F0.5  |
> | -------- | ----- | ----- | ----- |
> | 0.5      | 55.96 | 41.35 | 52.27 |
> | 0.8      | 55.65 | 41.35 | 52.05 |
> | 1.0      | 57.96 | 39.00 | **52.83** |
> | 1.2      | 56.75 | 40.93 | 52.68 |
> | 2.0      | 55.88 | 41.30 | 52.20 |
>
> **Table 4**: Results of various $\beta$ on BEA-train.
>
> | $\beta$ | P     | R     | F0.5  |
> | ------- | ----- | ----- | ----- |
> | 0.5     | 57.56 | 38.51 | 52.38 |
> | 1.0     | 57.96 | 39.00 | **52.83** |
> | 2.0     | 56.69 | 40.59 | 52.52 |
>
> As shown in Appendix A.1, the **optimal** hyperparameter setting is $\alpha=1.0$ and $\beta=1.0$.
>
> ---
>
> **Q6**: How to compute P_avg?
>
> **A6**: Actually, $P_{avg}$ is computed solely based on the target sentences and is **not** influenced by the source sentences. Therefore, the computation of $P_{avg}$ is straightforward.
>
> ---
>
> **Q7**: Why does MixEdit get its name?
>
> **A7**: "Mix" is so named as the augmented grammatical errors are **sampled from the Error Pattern Pool**, as depicted in Figure 1 of our paper. The term "Edit" is used to describe the **unit** of our proposed data augmentation method. We have coined the name "MixEdit" for our proposed method to **reflect the augmentation process**.
>
> ---
>
> **Q8**: Table 4 should be presented as a graph, to better illustrate the effectiveness of two proposed metrics.
>
> **A8**: We did consider presenting the results as a graph, but it was more feasible to showcase the **trade-off between Precision and Recall** in tabular form.
>
> ---
>
> **Q9**: How to acquire subwords by bype-pair-encoding?
>
> **A9**: The subwords are obtained using a off-the-shelf tokenizer following previous study [1], ensuring the **comparability** of our experiments to theirs.
>
> ---
>
> **Reference**
>
> [1] SynGEC: Syntax-Enhanced Grammatical Error Correction with a Tailored GEC-Oriented Parser [EMNLP 2022]

---

### Official Review · Reviewer_g98u · 2023-08-05

**Soundness:** 3

**Excitement:**

3: Ambivalent: It has merits (e.g., it reports state-of-the-art results, the idea is nice), but there are key weaknesses (e.g., it describes incremental work), and it can significantly benefit from another round of revision. However, I won't object to accepting it if my co-reviewers champion it.

**Paper Topic And Main Contributions:**

This paper aims to determine the optimal conditions for when grammatical error correction (GEC) systems benefit the most from data augmentation methods and attempts to develop a more effective augmentation approach based on the identified characteristics of effective augmentations. The contributions of this paper are as follows:

1) The paper introduces novel analysis perspectives for GEC data augmentation. Two interpretable metrics, Affinity, and Diversity, are proposed to uncover the underlying mechanisms of previous augmentation methods. Affinity measures the KL divergence, while Diversity quantifies the entropy of probabilistic distributions.
2) Through empirical exploration, the paper identifies the key characteristics of an effective GEC data augmentation strategy. Based on these findings, the authors introduce a new augmentation approach called MixEdit. This approach optimizes the model using more appropriate pseudo data and objectives.

The experiments conducted on English and Chinese benchmarks demonstrate that MixEdit is an outstanding method, even without additional data. Furthermore, it can be seamlessly integrated with existing augmentation approaches.


**Questions For The Authors:**

- A. Regarding footnote$^2$ on Page 4, as it differs from previous research, could you provide additional evidence or data to support the claim that a single augmented sample performs as effectively as multiple samples?


**Reasons To Accept:**

- **Motivation**: I greatly appreciate the motivation behind this work, as it aims to understand the underlying mechanisms of data augmentation approaches in the field of GEC. The paper provides valuable insights by integrating the strengths of previous methods and presents an interesting finding: an effective augmentation method will generate data with high Affinity and appropriate Diversity.

- **Novel Metrics**: The introduction of the proposed metrics (Affinity and Diversity) offers a statistical perspective to explain existing GEC data augmentation methods. The paper also justifies why these dimensions are crucial in GEC data augmentation.

- **Experiments**: The experiments conducted are solid and effectively demonstrate the efficacy of MixEdit across different languages (English, Chinese). Although existing augmentation methods may not significantly benefit from MixEdit, it proves to be highly effective in on-the-fly augmentation scenarios.


**Reasons To Reject:**

- **Soundness Problems**: The main argument throughout the paper suggests that a better augmentation method might produce data with high Affinity and appropriate Diversity, which serves as the motivation for introducing the MixEdit method (Line 261). However, I still have some concerns regarding the validity of this argument:

  1. While Table 4 presents the observed trend and Section 4.3 discusses these findings, it is essential to prove a more concrete relationship between Affinity and performance. Utilizing statistical measures like Pearson's correlation coefficient could strengthen the argument. Moreover, the trade-off of Diversity seems to be hard to examine in a more scientific way.

  2. In addition to the BEA-train dataset, it would be beneficial to explore other realistic datasets to analyze the relationship between Affinity/Diversity and augmentation performance.

- **Inconsistent logic of paper**: The abstract and initial sections of the paper indicate that the analysis of data augmentation mechanisms using the proposed metrics will be presented first, followed by the introduction of a novel augmentation method based on the findings. However, the subsequent sections deviate from this logical order by presenting the method and experimental results first (Section 4.1, 4.2) before revealing observations related to the two metrics (Section 4.3). This inconsistency in the paper's structure creates confusion for readers.


**Reproducibility:**

3: Could reproduce the results with some difficulty. The settings of parameters are underspecified or subjectively determined; the training/evaluation data are not widely available.

**Reviewer Confidence:**

4: Quite sure. I tried to check the important points carefully. It's unlikely, though conceivable, that I missed something that should affect my ratings.

**Typos Grammar Style And Presentation Improvements:**

- Line 224: datasets respectively. -> datasets, respectively.
- Line 241: axis -> perspective
- Line 283: Inspired by (Shen et al., 2020; Chen et al., 2021), -> Inspired by Shen et al., (2020) and Chen et al. (2021),
- Line 556: Following (Zhang et al., 2022a), -> Following Zhang et al. (2022a),

---

> ### Author Rebuttal · Authors · 2023-08-28
>
> We are grateful for your positive comments and insightful reviews. Please find below our point-by-point responses:
>
> ---
>
> **Q1**: It is essential to prove a more concrete relationship between Affinity and performance.
>
> **A1**: We agree with your suggestion as it is technically sound. To further confirm the relationship between Affinity/Diversity and GEC performance, we have **conducted additional experiments on other datasets** (English CLang8 and Chinese HSK).
>
> **Table 1**: Affinity and Diversity of data augmentation methods on **English CLang8**. The experimental settings are identical to those mentioned in Section 4.3 of our paper, with the only difference being the choice of datasets.
>
> | Method        | Affinity | Diversity | P     | R     | F0.5  |
> | ------------- | -------- | --------- | ----- | ----- | ----- |
> | Baseline      | -        | 8.81      | 63.04 | 44.69 | 58.25 |
> | DN            | 0.33     | **11.31**     | 28.52 | 24.48 | 27.60 |
> | PN Round=1    | 3.74     | 7.63      | 62.66 | 28.32 | 50.43 |
> | PN Round=2    | 3.55     | 7.63      | 60.24 | 34.12 | 52.24 |
> | PN Round=4    | 3.17     | 8.05      | 56.70 | 38.58 | 51.83 |
> | PN Round=8    | 2.65     | 8.47      | 49.70 | 43.36 | 48.29 |
> | PN Round=16   | 1.92     | 9.10      | 40.59 | 48.21 | 41.92 |
> | BT $\beta=0$  | 1.04     | 7.04      | 57.50 | 13.96 | 35.41 |
> | BT $\beta=3$  | 3.59     | 7.60      | 62.63 | 38.94 | 55.83 |
> | BT $\beta=6$  | 3.73     | 8.15      | 57.73 | 46.71 | 55.13 |
> | BT $\beta=9$  | 3.08     | 8.55      | 53.07 | 51.19 | 52.68 |
> | BT $\beta=12$ | 2.55     | 8.87      | 49.38 | **52.63** | 50.00 |
> | RT            | 0.53     | 11.14     | 18.43 | 37.16 | 20.50 |
> | MixEdit       | **19.29**    | 8.85      | **64.50** | 38.32 | **56.75** |
>
> **Table 2**: Affinity and Diversity of data augmentation methods on **Chinese HSK**.
>
> | Method        | Affinity | Diversity | P     | R     | F0.5  |
> | ------------- | -------- | --------- | ----- | ----- | ----- |
> | Baseline      | -        | 8.78      | 49.10 | 25.30 | 41.33 |
> | DN            | 0.39     | 10.10     | 28.84 | 26.56 | 28.35 |
> | PN Round=1    | 3.07     | 6.40      | 45.84 | 8.86  | 24.98 |
> | PN Round=2    | 2.73     | 6.59      | 43.54 | 13.28 | 19.91 |
> | PN Round=4    | 2.25     | 6.91      | 44.96 | 15.12 | 32.24 |
> | PN Round=8    | 1.43     | 7.43      | 42.83 | 16.89 | 32.76 |
> | PN Round=16   | 1.17     | 8.20      | 39.82 | 21.24 | 33.89 |
> | BT $\beta=0$  | 1.30     | 5.44      | 27.07 | 3.34  | 11.18 |
> | BT $\beta=3$  | 3.00     | 8.24      | 45.48 | 15.78 | 33.04 |
> | BT $\beta=6$  | 2.73     | 8.10      | 41.72 | 26.32 | **37.35** |
> | BT $\beta=9$  | 2.17     | 8.8.3     | 37.83 | 30.27 | 36.03 |
> | BT $\beta=12$ | 1.75     | 9.45      | 32.27 | **32.59** | 32.33 |
> | RT            | 0.46     | **10.57**     | 19.39 | 29.84 | 20.84 |
> | MixEdit       | **3.95**     | 8.52      | **47.81** | 18.31 | 36.15 |
>
> We compute the **Pearson's correlation coefficient** for the 5 GEC data augmentation methods involved: DN, PN, BT, RT and MixEdit. For PN and BT, we choose the hyper-parameter configurations that yield the highest F0.5 scores. The Pearson's correlation coefficients between F0.5 and Affinity are **0.9485 on BEA-train, 0.6239 on CLang8, and 0.7717 on HSK**. The results indicate a **strong or moderate correlation between Affinity and F0.5**.
>
> ---
>
> **Q2**: The trade-off of Diversity seems to be hard to examine in a more scientific way.
>
> **A2**: As stated in Section 4.3 of our paper, We **qualitatively** analyze the effect of Diversity on GEC performance within a fixed method. The trade-off of Diversity is apparent when adjusting certain hyper-parameters responsible for Diversity (Round for PN and $\beta$ for BT). As Diversity increases, Precision and F0.5 initially increase and then decrease, reaching their peak at an appropriate setting. Meanwhile, Recall continues to increase.
>
> ---
>
> **Q3**: How does the relationship between Affinity/Diversity and performance show in other datasets?
>
> **A3**: To explore the relationship between Affinity/Diversity and performance across different datasets, we have conducted additional experiments on English CLang8 and Chinese HSK. The results and analyses are provided above. The relationship between Affinity/Diversity and performance is **similar to the results on BEA-train shown in our paper**.
>
> ---
>
> **Q4**: Inconsistent logic of paper. Section 4.3 should be presented before Section 4.1 and 4.2.
>
> **A4**: The suggestion to rearrange the narrative order of Section 4 is valuable and we will make the necessary adjustments.
>
> ---
>
> **Q5**: Could you provide additional evidence or data to support the claim that a single augmented sample performs as effectively as multiple samples?
>
> **A5**: In our preliminary experiments, we found using N=2 augmented samples can improve F0.5 scores by **less than 0.1** compared with N=1, and sometimes even have a negative impact. We speculate that too many augmented samples may dilute the valuable supervision information obtained from realistic sources. Additionally, increasing N would **significantly escalate the training cost**. For instance, with N=2, it would entail **three** feed forwards and back propagations for a single sample (two for augmented samples and one for the realistic sample). We will delve further into this issue in the final version of our work.

---

### Official Review · Reviewer_ytRu · 2023-08-05

**Soundness:** 4

**Excitement:**

4: Strong: This paper deepens the understanding of some phenomenon or lowers the barriers to an existing research direction.

**Paper Topic And Main Contributions:**

This paper presents two efficiently computable and interpretable metrics that provide a better understanding of data augmentation in Grammar Error Correction (GEC). Previous studies have suggested that the similarity between real and pseudo data, as well as the diversity within pseudo data, play key roles in GEC augmentation. However, there's been no quantitative measure to evaluate the relationship between these perspectives and the performance of various data improvement methods in a coherent experimental setting. This paper offers a solution to this by proposing a method for measuring affinity using the inverse function of KL divergence and diversity in terms of entropy.

The findings using these measurements indicate that affinity has a direct correlation with performance. As for diversity, it presents a trade-off between precision and recall, according to the authors.

Based on this analysis, the authors designed a loss function, named MixEdit, that appropriately mixes the losses generated from each of the real and augmented data using KL divergence loss and cross-entropy loss, achieving performance comparable to state-of-the-art models.


**Reasons To Accept:**

- The paper quantifies the known insights into GEC's data augmentation by demonstrating its relationship with performance.

- It also showed that state-of-the-art performance can be achieved by designing a learning objective function to control these quantities.

- Through systematic experiments comparing multiple data augmentation techniques, as well as detailed experiments including ablation studies, the paper revealed that the affinity with real data and the use of an error pattern pool have significant effects on the GEC performance.


**Reasons To Reject:**

While there is no major reason to reject the paper, the reader's understanding of the results may be enhanced if the interpretation of the results is discussed concerning substantially comparable settings, especially with respect to the numerous experimental settings listed in Table 3.
Although the efficiency of sampling is mentioned in the introduction, it does not seem to be revisited later in the paper. If this is indeed the case, the claim needs to be revised.


**Reproducibility:**

3: Could reproduce the results with some difficulty. The settings of parameters are underspecified or subjectively determined; the training/evaluation data are not widely available.

**Reviewer Confidence:**

3: Pretty sure, but there's a chance I missed something. Although I have a good feel for this area in general, I did not carefully check the paper's details, e.g., the math, experimental design, or novelty.

---

> ### Author Rebuttal · Authors · 2023-08-28
>
> Thanks for your appreciation of our work's contributions. We are honored to have the opportunity to discuss with you.
>
> ---
>
> **Q1**: Could you elaborate the experiment results concerning comparable settings, especially with respect to the numerous experimental settings listed in Table 3?
>
> **A1**: Table 3 reports previous studies' results with different experimental settings, which make it hard to compare fairly with each other due to the selection of backbones, datasets and training procedure. However, the results of **SynGEC** [1] and our experiments are **comparable fairly**. We will clarify the difference of experimental settings and improve the discussion on the results in Table 3 in our final version.
>
> ---
>
> **Q2**: Discussion regarding the sample efficiency should be added.
>
> **A2**: The comparison of various augmentation methods' sample efficiency is shown in Table 4 of our paper. With the same number of augmented samples, MixEdit can achieve the highest F0.5 score on BEA-train. To further confirm our finding, we have **conducted additional experiments on other datasets** (English CLang8 and Chinese HSK).
>
> **Table 1**: Affinity and Diversity of data augmentation methods on **English CLang8**. The experimental settings are identical to those mentioned in Section 4.3 of our paper, with the only difference being the choice of datasets.
>
> | Method        | Affinity | Diversity | P     | R     | F0.5  |
> | ------------- | -------- | --------- | ----- | ----- | ----- |
> | Baseline      | -        | 8.81      | 63.04 | 44.69 | 58.25 |
> | DN            | 0.33     | **11.31**     | 28.52 | 24.48 | 27.60 |
> | PN Round=1    | 3.74     | 7.63      | 62.66 | 28.32 | 50.43 |
> | PN Round=2    | 3.55     | 7.63      | 60.24 | 34.12 | 52.24 |
> | PN Round=4    | 3.17     | 8.05      | 56.70 | 38.58 | 51.83 |
> | PN Round=8    | 2.65     | 8.47      | 49.70 | 43.36 | 48.29 |
> | PN Round=16   | 1.92     | 9.10      | 40.59 | 48.21 | 41.92 |
> | BT $\beta=0$  | 1.04     | 7.04      | 57.50 | 13.96 | 35.41 |
> | BT $\beta=3$  | 3.59     | 7.60      | 62.63 | 38.94 | 55.83 |
> | BT $\beta=6$  | 3.73     | 8.15      | 57.73 | 46.71 | 55.13 |
> | BT $\beta=9$  | 3.08     | 8.55      | 53.07 | 51.19 | 52.68 |
> | BT $\beta=12$ | 2.55     | 8.87      | 49.38 | **52.63** | 50.00 |
> | RT            | 0.53     | 11.14     | 18.43 | 37.16 | 20.50 |
> | MixEdit       | **19.29**    | 8.85      | **64.50** | 38.32 | **56.75** |
>
> **Table 2**: Affinity and Diversity of data augmentation methods on **Chinese HSK**.
>
> | Method        | Affinity | Diversity | P     | R     | F0.5  |
> | ------------- | -------- | --------- | ----- | ----- | ----- |
> | Baseline      | -        | 8.78      | 49.10 | 25.30 | 41.33 |
> | DN            | 0.39     | 10.10     | 28.84 | 26.56 | 28.35 |
> | PN Round=1    | 3.07     | 6.40      | 45.84 | 8.86  | 24.98 |
> | PN Round=2    | 2.73     | 6.59      | 43.54 | 13.28 | 19.91 |
> | PN Round=4    | 2.25     | 6.91      | 44.96 | 15.12 | 32.24 |
> | PN Round=8    | 1.43     | 7.43      | 42.83 | 16.89 | 32.76 |
> | PN Round=16   | 1.17     | 8.20      | 39.82 | 21.24 | 33.89 |
> | BT $\beta=0$  | 1.30     | 5.44      | 27.07 | 3.34  | 11.18 |
> | BT $\beta=3$  | 3.00     | 8.24      | 45.48 | 15.78 | 33.04 |
> | BT $\beta=6$  | 2.73     | 8.10      | 41.72 | 26.32 | **37.35** |
> | BT $\beta=9$  | 2.17     | 8.8.3     | 37.83 | 30.27 | 36.03 |
> | BT $\beta=12$ | 1.75     | 9.45      | 32.27 | **32.59** | 32.33 |
> | RT            | 0.46     | **10.57**     | 19.39 | 29.84 | 20.84 |
> | MixEdit       | **3.95**     | 8.52      | **47.81** | 18.31 | 36.15 |
>
> The results reveal that **MixEdit achieves the highest F0.5 score on CLang8 and comparable F0.5 on HSK**. Surprisingly, backtranslation (BT) achieves the highest F0.5 on Chinese HSK. We speculate that Chinese grammatical errors are inherently more intricate, providing an advantage for BT as it can generate pseudo grammatical errors that are closer to authentic ones. Nonetheless, it is worth noting that BT relies on an **additional model** to generate grammatical errors, which introduces **efficiency concerns**.
>
> ---
>
> **Reference**
>
> [1] SynGEC: Syntax-Enhanced Grammatical Error Correction with a Tailored GEC-Oriented Parser [EMNLP 2022]

---

### Meta-Review · Area_Chair_9xR4 · 2023-09-21

**Recommendation:** 3

**Metareview:**

The paper addresses the topic of data augmentation for Grammatical Error Correction (GEC). They introduce two measurements of data quality – Affinity (based on KL-divergence) and Diversity (measured in terms of entropy). They propose a MixEdit loss function that optimizes the model using more appropriate pseudo data based on the two metrics.

Experiments on English and Chinese benchmarks show that high Affinity seems to improve model performance, but there is no consistent trend for the Diversity metric performance-wise. They conclude that a data augmentation strategy should be characterized by high Affinity and appropriate Diversity.

The paper could be substantially improved by strengthening  the argument regarding the utility of the proposed metrics. Specifically, it  is not clear  how to quantify “appropriate diversity” for a dataset. The paper should make a better effort to explore this issue and to make a more convincing argument regarding the utility of the metric. The individual reviews have specific suggestions about this. The paper would also be strengthened significantly if experiments on other languages were added.

---

### Decision · Program_Chairs · 2023-10-07

**Decision:**

Accept-Findings

**Comment:**

The paper addresses the topic of data augmentation for Grammatical Error Correction (GEC). They introduce two measurements of data quality – Affinity (based on KL-divergence) and Diversity (measured in terms of entropy). They propose a MixEdit loss function that optimizes the model using more appropriate pseudo data based on the two metrics.

Experiments on English and Chinese benchmarks show that high Affinity seems to improve model performance, but there is no consistent trend for the Diversity metric performance-wise. They conclude that a data augmentation strategy should be characterized by high Affinity and appropriate Diversity.

The paper could be substantially improved by strengthening  the argument regarding the utility of the proposed metrics. Specifically, it  is not clear  how to quantify “appropriate diversity” for a dataset. The paper should make a better effort to explore this issue and to make a more convincing argument regarding the utility of the metric. The individual reviews have specific suggestions about this. The paper would also be strengthened significantly if experiments on other languages were added.